# Insect Insights at the Single-Cell Level: Technologies and Applications

**DOI:** 10.3390/cells13010091

**Published:** 2023-12-31

**Authors:** Chao Sun, Yongqi Shao, Junaid Iqbal

**Affiliations:** 1Analysis Center of Agrobiology and Environmental Sciences, Zhejiang University, Hangzhou 310058, China; chaosun@zju.edu.cn; 2Institute of Sericulture and Apiculture, College of Animal Sciences, Zhejiang University, Hangzhou 310058, China

**Keywords:** scRNA-seq, single cell technology, insect

## Abstract

Single-cell techniques are a promising way to unravel the complexity and heterogeneity of transcripts at the cellular level and to reveal the composition of different cell types and functions in a tissue or organ. In recent years, advances in single-cell RNA sequencing (scRNA-seq) have further changed our view of biological systems. The application of scRNA-seq in insects enables the comprehensive characterization of both common and rare cell types and cell states, the discovery of new cell types, and revealing how cell types relate to each other. The recent application of scRNA-seq techniques to insect tissues has led to a number of exciting discoveries. Here we provide an overview of scRNA-seq and its application in insect research, focusing on biological applications, current challenges, and future opportunities to make new discoveries with scRNA-seq in insects.

## 1. Introduction

Transcriptomic analysis is an effective tool for investigating gene expression patterns, allowing researchers to obtain significant insights into the activity of groups of cells or tissues. It has been widely employed in biological research since next generation sequencing (NGS) became available to the public in 2005 [1]. In the past decade, the majority of the transcriptomic research has come through bulk RNA-seq which offers a comprehensive view of gene expression across the entire transcriptome, providing averaged data from an entire tissue or a large population of individual cells. Thus, information from bulk RNA-seq is actually from a mixture of tissues which can hide important differences between closely related cell types within a tissue.

In 2009, a major breakthrough in molecular biology led to the development of the first single-cell RNA sequencing (scRNA-seq) [2], which was named as the Method of the Year 2013 by “Nature Methods” [3]. Compared with traditional sequencing technology, single-cell technologies have the advantages of revealing heterogeneity among individual cells [4]. This revolutionary technique is the foundation for exploring the genetic makeup of individual cells in a new way, offering remarkable insights into their distinct gene expression patterns. By single-cell omics, we can identify rare and essential cell types that play an important role in a dominant sub-population in a biological process such as organ development [5].

Over the past decade, there has been a remarkable rise in various scRNA-seq technologies [6,7,8,9,10,11,12]. This technology has been widely used in medical science to identify rare cell types or intermediate cells at the single-cell level, to analyze the biological development process of complex cell lineages or embryos, and to explore the pathogenesis of cancer (Reviewed by [13]).

If we look back upon the development history of high-throughput sequencing in the past ten years, it is not difficult to find that advanced and mature technology applied in medicine will be introduced to entomology gradually, thereby promoting the development of this field. In recent years, the application of single-cell transcriptomics to insect tissues has led to a number of exciting discoveries, such as uncovering new cell types [14,15,16], identifying gene regulatory mechanisms [17,18], revealing random allelic gene expression [19,20] and investigating developmental processes dynamics [21,22,23,24].

In this review, we will provide a summary of recent single-cell transcriptome studies with an emphasis on those that have been used in insects, focusing on advancement of biological applications. Current challenges and future opportunities for making new discoveries using scRNA-seq in insects will also be discussed. For a broader view of single-cell analysis beyond scRNA-seq in insect research, see [6,7,8,11,25].

## 2. The Effectiveness and Overview of the Single-Cell RNA-Sequencing Workflow

Following the introduction of the initial scRNA-seq method [2], many scRNA-seq methods have been proposed for scRNA-seq studies [26]. scRNA-seq is a powerful technology that enables the determination of precise gene expression patterns from tens of thousands of cells. This technique has the potential to reveal underlying mechanisms of gene regulation and identify cell types and functions, providing deeper insight into developmental processes in diverse biological samples. scRNA-seq is revolutionizing our fundamental understanding of biology, and this technique has opened new areas of research beyond descriptive studies of cell states. Potential applications of scRNA-seq include identifying genes involved in stem cell regulatory networks and providing new insights into physiological structure-function relationships in various tissues and organs (Reviewed by [13]). As the availability of standardized bioinformatics pipelines improves, this work will provide new insights into biological systems and create new opportunities for biological science [9].

Depending on research goal or biological system, a variety of scRNA-seq protocols have been used. Generally, these steps can be categorized into five major ones (Figure 1): (1) Tissue dissection, (2) single-cell suspension, (3) single-cell capture, (4) cDNA synthesis and library construction, and (5) sequencing and data analysis. The actual process of scRNA-seq starts by making a single-cell suspension. Tissues are gently broken down to release a single cell by using a specific set of enzymes or mechanical forces, or both. This step is critical because it ensures that each cell remains distinguishable and allows accurate profiling of its unique genetic activity. After preparing a single-cell suspension, the next step is single cell capture. Individual cells and their contents are separated into compartments or droplets and handled separately. There are several different ways to do this: Fluorescence-activated cell sorting (FACS), microfluidics, laser capture microdissection, micromanipulation or micropipetting, and limiting dilution, each with its own set of advantages. In the methods mentioned above, the most frequently employed approaches for single-cell capture are the FACS-based and microfluidics-based methods. The other approaches are time-consuming or low-throughput and are not frequently used in scRNA-seq research. To date, various scRNA-seq platforms have been established, such as CEL-seq [27], Smart-seq2 [28], MARS-seq [29], inDrop [30], Drop-seq [31], 10× Genomics [32], Sci-RNA-seq [33], MATQ-seq [34], SPLIT-seq [35], and SEQ-well [36]. Distinctive features among these techniques include the use of a unique molecular identifier (UMI), cDNA coverage (full-length or 5′/3′), utilization of a plate or droplet-based platform, throughput, targeted read depth, and cost consideration. In this review, the general workflow of scRNA-seq and its various platforms have been briefly described; however, for a more detailed discussion of the similarities and differences among the different platforms used, see review [37,38,39]. In the mentioned techniques, only four have been used in insect scRNA-seq studies: plate-based Smart-seq2 and droplet based inDrop, Drop-seq, and 10× Genomics (Table 1). 10× Genomics stands out as the preferred choice for research related to insects, primarily owing to its exceptional accessibility, superior data quality, and unparalleled platform stability [40].

During the analysis phase, it is crucial to perform a second round of quality control using appropriate tools. In the single-cell context, Seurat [41], scran [42] and scanpy [43] are best suited for quality control. Seurat is the most widely used tool, as evidenced by a large number of citations, and has built-in functions for filtering of poor-quality cells. Essentially, the following quality control indicators should be used to assess whether a cell should be retained in the analysis: the number of genes, the number of UMI (transcripts), and the percentage of mitochondrial genes and the percentage of ribosomal protein genes in each cell. These indicators serve as important criteria to assess and ensure the integrity and reliability of the single-cell data. There is no fixed standard for setting filter thresholds; they usually depend on the type of cell and tissue. Yu et al. filtered out cells with less than 990 or more than 4200 expressed genes, and with an unusually high number of UMIs (>37,000) or mitochondrial gene percentage (>25%) [44]. Sun et al. used the following criteria to filter out low-quality cells: (i) gene counts > 3000 per cell; (ii) UMI counts > 12,000 per cell; and (iii) percentage of mitochondrial genes > 30% [45]. Adjusting the above QC threshold flexibly according to the specific disease state and the diversity of tissue types is recommended. For a more in-depth understanding of the quality control process, see the detailed insights provided in the review [46].

## 3. Single-Cell Transcriptomics in Insects: Applications

### 3.1. General Overview

Insects are a good model for scRNA-seq studies because of their short life span, distinct developmental stages, and ease of handling. A few years ago, scientist started using single-cell sequencing (scRNA seq) to study insects. One of the very first studies to apply scRNA seq to insects was conducted by Karaiskos and his colleagues in 2017 [47]. They reported that a *Drosophila* embryo, consisting of approximately 6000 cells, revealed a new mechanism underlying the embryo pattern formation. They further identified that the spatial regulation of Hippo signaling in the early stages of embryonic development indicates a mechanism enabling cells to grow at different rates rather than synchronously. Since then, scientists have made a great progress in insect research using scRNA-seq (Table 1). Interestingly, this technology has been used not only for the model insect *Drosophila*, but also for other insects, and numerous insect organs have been studied (Figure 2, Table 1). Recently, researchers have provided in-depth insights into the mechanisms governing embryonic development [47,48], germ cell development [44,45,49,50], brain aging [51], intestinal stem cell differentiation [52], tumorigenesis [53,54], immune cell specification [55,56], and neural development [57,58,59] of insects, using advanced techniques such as scRNA-seq In addition, scientists have used scRNA-seq to compare different caste members in social insects, unravelling the mysteries of their evolution [60,61,62].

In the next sections, we will talk about how scientists are using scRNA-seq in insects. This technology enables the study of genes in single cells and has opened up new research opportunities that allow us to explore the reproductive, immune, digestive, and nervous systems of insects with unprecedented resolution. We will explore how scRNA-seq has been used to investigate the molecular mechanisms of these vital biological systems. By examining each of these areas, we are gaining insight into not only insect physiology, but also their evolution, adaptation, and ecological importance.

**Table 1 cells-13-00091-t001:** Overview of various insect species, the specific tissues or organs studied, developmental stages, the employed single-cell RNA sequencing (scRNA-seq) technology, and the primary research directions.

Insects	Tissue	Stage	Sequencing Technology	Research Direction	Reference
*Drosophila melanogaster*	Brain	Pupae	Smart-seq2	Olfactory projection neuron	[63]
*D. melanogaster*	Mid brain	Adult	Drop-Seq	Cellular diversity	[64]
*D. melanogaster*	Brain	Different days old adults	10× Genomics	Diversity of cell types during aging	[51]
*D. melanogaster*	Brain (normal/starved)	Larvae	10×	Cell diversity	[65]
*D. melanogaster*	Central Nervous System, Brain, Ventral Nerve Cord	1, 24, 48 and 96 h ALH	10×	Developmental biology and neuroscience	[66]
*D. melanogaster*	Brain	3rd instar larvae	10×	Neuron development	[67]
*D. melanogaster*	Brain	Pupae	10×	Neuronal differentiation	[68]
*D. melanogaster*	Brain	Larvae	Drop-seq	Tyrosine kinase pathway	[69]
*D. melanogaster*	Brain	3rd instar larvae (m + f)	10×	Neurogenesis	[70]
*D. melanogaster*	Brain	Adults	10×	Behavioral changes	[71]
*D. melanogaster*	Brain	Adults	10×	Neurobiology	[72]
*D. melanogaster*	Brain	Adults	10×, CELL-seq 2	Circadian and dopaminergic neurons	[73]
*D. melanogaster*	Brain	(0–6 h APF, 24–30 h APF, 48–54 h APF, and 1–5 day adults).	Smart-seq 2	Olfactory	[74]
*D. melanogaster*	Optic lobes	Pupae	10×	Neuronal transcriptomes	[75]
*D. melanogaster*	Optic lobes	Pupae	10×	Neuronal diversity, brain development across species	[76]
*D. melanogaster*	Optic lobe	Adults	Drop-seq	Development and function of brain	[77]
*D. melanogaster*	Optic lobe	3rd instar larvae (male + female)	10×	Neuronal diversity	[78]
*D. melanogaster*	Ventral nerve cord	Adults	10×	neuroblast tumors	[54]
*D. melanogaster*	Ventral nerve cord	Adults (male + female)	10×	Neuro development and behavior	[59]
*D. melanogaster*	Nervous system (Brain ventral nerve cord)	1 h, 24 h, 48 h, or 96 h after larval hatching	10×	Neural developmental	[57]
*D. melanogaster*	Pupal CNS	Pupae (male + female)	10×	Reproductive behaviors	[79]
*D. melanogaster*	larval CNS	1, 24 and 48 h after larval hatching	10×	Development	[58]
*D. melanogaster*	Stored cells (T4/T5 neurons)	Pupae	10×	Neuronal connectivity	[80]
*D. melanogaster*	lamina neurons	Pupae	10×	Cell-type specific expression of wiring genes	[81]
*D. melanogaster*	Blood	Larvae	10×	Cellular heterogeneity	[82]
*D. melanogaster*	Blood	Larvae	inDrop; 10×	Diversity of hemocytes	[83]
*D. melanogaster*	Hemocyte	Larvae	10×	Insect immunity	[55]
*D. melanogaster*	Hemocyte	Larvae	10×	Immunity (evolution)	[56]
*D. melanogaster*	Ovary	Adults	10×	Cell types and subtypes	[84]
*D. melanogaster*	Ovary	Adults	10×	Stem cells	[85]
*D. melanogaster*	Ovary	Virgin female adults	10×	Germ stem cell research	[45]
*D. melanogaster*	Ovary	Adults	10×	Tumorigenesis	[53]
*D. melanogaster*	Ovary	Adults	10×	Cell atlas of the adult *Drosophila* ovary	[86]
*D. melanogaster*	Ovary	Adults	10×	Oogenesis	[87]
*D. melanogaster*	Ovary	Larvae	10x	Ovary morphogenesis	[49]
*D. melanogaster*	Gut	Adults	10×	Enteroendocrine cells	[88]
*D. melanogaster*	Midgut	7-day old adults (female)	inDrop and 10×	Gene function	[89]
*D. melanogaster*	Midgut	Adults (7, 30 and 60 days old)	10×	Age related tissue homeostasis, intestinal stem cells	[52]
*D. melanogaster*	eye disc	Larvae	Drop-seq	Apoptosis due to mutation	[90]
*D. melanogaster*	eye	Adults	10×	Visual system	[91]
*D. melanogaster*	Eye-antennal discs	3rd instar larvae	10×	Neuronal differentiation	[92]
*D. melanogaster*	Antenna	Mid pupal stage	Smart-seq 2	Olfactory system	[93]
*D. melanogaster*	3rd antennal segment	Adults	Smart-seq 2	Development of olfactory system	[94]
*D. melanogaster*	Testis	Males	Smart-seq 2	Spermatogenesis	[44]
*D. melanogaster*	Testis	2–3 days old males	10×	Male fertility + spermatid elongation	[95]
*D. melanogaster*	Testis	One day old males	10×	Wolbachia affected germ cells	[50]
*D. melanogaster*	Testis	Young and old males	10×	Mutational signatures of the *Drosophila* ageing germline	[96]
*D. melanogaster*	Testis	Males	10×	Germline	[97]
*D. melanogaster*	Testis	Adults	10×	Spermatogenesis	[98]
*D. melanogaster*	multiple	Adults (male + female)	10× and Smart-seq2	Cell Atlas	[99]
*D. melanogaster*	Lymph gland	Larvae	Drop seq	Hemocyte development and cellular immunity	[100]
*D. melanogaster*	Foreleg	Pupae (pupal development)	10×	Sensory organs	[101]
*D. melanogaster*	Leg disc	Larvae	10×	Leg development	[102]
*D. melanogaster*	Embryo	Embryos	Drop-seq	Virtual embryo	[47]
*D. melanogaster*	Embryo	Embryos	Drop-seq	Embryonic development	[48]
*D. melanogaster*	Abdominal cuticle (adipose tissue, oenocytes, and abdominal muscles)	Adults	10×	Homeostasis	[103]
*D. melanogaster*	Wing disc	Larvae	Drop-seq	Developmental processes	[104]
*D. melanogaster*	Wing disc	3rd instar Larvae	Drop-seq	Muscle diversification	[105]
*D. melanogaster*	Wing disc	Pupae	10×	Cell–cell interactions between epithelial and myogenic cells.	[106]
*Bombyx mori* (Silkworm)	Silk gland	Larvae	10×	Silk gland cells and their gene expression dynamics,	[107]
*Bombyx mori* (Silkworm)	Hemocytes	Larvae	10×	Cellular characteristics upon virus infection	[108]
*Aedes aegypti* (Mosquito)	Midgut	Adults (female)	10×	Heterogeneity of midgut cells, Cellular homeostasis, nutritional absorption	[109]
*Anopheles gambiae* (Mosquito)	Hemocytes	Adults (female)	10×	Mosquito immunity to malaria infection	[110]
*Harpegnathos saltator* (Ant)	Brain	Adults (worker and gamergates)	10×	Caste-specific cellular plasticity	[60]
*Monomorium pharaonis* (Ant)	Brain	Queens, gynes (virgin queens), workers and males	Drop-seq	Caste/sex-specific changes in cell types (behavior)	[61]
*Apis mellifera* (Honey bee)	Brain	Adults (soldiers and foragers)	10×	Relationship between genotypic variation and phenotypic variation in collective behavior.	[62]
*Dalotia coriaria* (Rove beetle)	Tergal gland	Males	10×	Evolution	[111]

Overview of single-cell technologies used in insects. Abbreviation: *D. melanogaster, Drosophila melanogaster;* 10×, 10× genomics; Drop-seq, Droplets sequencing; SMART-seq, Switching mechanism at the 5’end of the RNA template; InDrops, Indexing droplets; CELL-seq, Cell expression by linear amplification and sequencing.

### 3.2. Application in the Insect Nervous System

Making sense of any complex system involves identifying constituent elements and understanding their individual functions and interactions. Nervous systems are no exceptions. A number of single-cell transcriptome studies have been conducted to explore the complexity of insect nervous systems (See Table 1). Unlike the peripheral nervous system and sympathetic nervous system, which have received negligible attention with few related publications [101], the majority of research on insect nervous systems is focused on the central nervous system (CNS).

Comprehensive transcriptomic atlases of the complete CNS have revealed the complex interplay between cell state and circuit function and behavior in *Drosophila* larvae [57,66] and revealed how group genetics can shape a collective phenotype by modulating individual brain gene regulatory network architecture in honeybee colonies [62]. Multiple progenitor subtypes across several *Drosophila* larval stages with differential gene expressions have also been identified to provide candidate genes as cell type specific markers and as having functional roles during development [58]. Excluding a few studies on the ventral nerve cord [59] which revealed more than 100 transcriptionally distinct cell types, and those on the entire central nervous system, most studies on the CNS are mainly focused on the brain.

#### 3.2.1. The Whole Brain

As an ideal subject for the application of transcriptional analysis, the *Drosophila* brain involves abundant cell- and tissue-specific genes that provide accurate information for identifying cell types through the use of scRNA-seq [48]. Avalos et al. [65] and Davie et al. [51] profiled the entire *Drosophila* brain of larvae and adults, respectively. Their searches revealed complex cell clusters in the *Drosophila* brain and provided databases to explore cell diversity and assess genetic profiles from developmental, functional, and behavioral perspectives. Further research on the *Drosophila* larval brain characterized the transcriptome landscape of thousands of type II neurons and their progenitors and revealed genes that vary during neural differentiation [70]. In ants, comprehensive profiling of whole-brain single-cell transcriptomes across the full panel of distinct adult phenotypes of different sexes, castes, and reproductive roles were obtained to map important aspects of multi-brain complementarity and functional coordination in a super organismal ant colony, with male brains having higher abundances of optic lobe neurons than workers. In addition, the transition from virgin to inseminated queens leads to changes in the frequency of cell types of about 35%, suggesting ongoing neurogenesis or programmed cell death during this role shift [61]. scRNA-seq of insect brain has also deepened our understanding of disease processes and stress factors. For example, in a study of tumorigenesis [54], the authors identified a subset of genes involved in the temporal patterning of larval neuroblasts (NBs) that are redeployed in tumors to generate a differentiation trajectory responsible for creating tumor cell heterogeneity. Similarly, a study on deformed wing virus (DWV-A), reported that glial cells are involved in the effects of DWV-A, instead of neurons [112]. Another study on the exposure of flies to fetal alcohol and its effects on gene expression in the brain [71] revealed that the genes associated with glutathione metabolism, lipid transport, glutamate and GABA metabolism, and vision, featuring in sexually dimorphic global multi-cluster interaction networks were regulated. The mentioned studies contribute to uncovering responsible transcriptional regulation or signaling pathways related to disease processes and stress factors. Furthermore, the present state of environmental pollution not only heightens the risk of diseases but also functions as a persistent stressor, impacting both our well-being and the planet [113,114]. In future, scRNA-seq should be extended to investigate various diseases and stress factors (environmental and others) for enhanced understanding.

#### 3.2.2. Olfactory and Visual System

To respond to their surroundings, insects rely on the input of a network of sensory organs. Much attention has focused on the antennae and eyes. Specific parts of the nervous system from different stages have also been profiled, including olfactory neurons and optic lobes. An scRNA-seq study retrieved 1000 pupal *Drosophila* olfactory projection neurons from a driver line labelling about 40 types from just three lineages, and resolved 30 clusters [63]. Xie et al. [74] also obtained the single-cell transcriptomes of *Drosophila* olfactory projection neurons at different developmental stages. They identified 21 transcriptomic clusters corresponding to 20 projection neuron types and discovered that projection neuron transcriptomes reflect unique biological processes unfolding at each stage. Using single-cell RNA sequencing, transcriptomic clusters and cell types of olfactory receptor neurons were identified as well [93]. Further transcriptomic research of olfactory receptor neurons in *Drosophila* at different life stages reflected axon trajectory choice in early development and sensory modality in adults and found that olfactory receptor neurons maintain expression of the same olfactory receptors across development [94]. Together, these studies uncovered cellular diversity of olfactory neurons and their diverse strategies and complex regulatory networks to coordinate physiology and connectivity.

The *Drosophila* visual system serves as an excellent model for studying complex behaviors and the molecular mechanisms underlying the development and evolution of the neural system. Over the past six years, the study of the *Drosophila* visual system, in particular the optic lobe, has made considerable progress due to large-scale transcriptomic single-cell studies [75,115]. *Drosophila* optic lobes exhibit moderate complexity, having approximately 200 cell types, yet can support complex behaviors [75,76]. Recently, several studies have extensively evaluated transcriptomic diversity in *Drosophila* optic lobes. Notably, two of the most recent studies on optic lobes examined transcriptomes during pupal development and achieved higher resolution, identifying around 200 clusters, in contrast to previous studies that identified fewer clusters [75,76,77]. Further, it is reported that neurons wrap around neuropils during development and die just before adulthood. In addition, different kinds of neurons play a role in the separation of dorsal and ventral visual circuits by utilizing different Wnt signaling across various stages of development [76]. Another study used scRNA-seq to examine how neurons in the fly visual system make precise connections. They found that nuclear receptor transcription factors (TFs) such as Hr3, Hr4 and ftz-f1 change their activity during the development of the visual system. They are part of a series of TFs activated by a hormone called ecdysone. These TFs, which include the ecdysone receptor EcR, are active in all neurons simultaneously, and their activity corresponds to a specific phase of development. They appear to regulate the expression of genes involved in how neurons are wired together, and this could be crucial to ensuring the right connections at the right time [81]. Furthermore, single-cell transcriptome data from over 27,000 adult *Drosophila* eye cells at different time points revealed different cell types and novel markers. In particular, differences between photoreceptor subtypes controlled by rhodopsin expression have been identified [91].

In conclusion, the use of scRNA-seq to investigate the visual system of insects, especially *Drosophila*, has shed light on the complex molecular underpinnings of neural development, evolution, and behavior. With its remarkable similarities to the vertebrate brain and moderate complexity, the *Drosophila* visual system has proven to be an important model for unraveling the mysteries of complex behaviors. Moreover, the combination of multi-omics methods and functional research will almost certainly uncover additional levels of complexity within these systems, opening up the possibility for novel treatment strategies and novel bioengineering applications.

### 3.3. Application in the Insect Digestive System

Unlike mammals, the digestive system in insects is relatively simple and can be divided into three main regions: the foregut, midgut, and hindgut. The midgut is the major site for digestion, whereas the foregut is mostly responsible for food storage, and the hindgut is primarily responsible for water reabsorption. The midgut of the insect (especially the fruit fly) is a remarkably regenerative organ that has been extensively utilized in recent years as a model system to examine the function of signaling pathways in coordinating stem cell proliferation and differentiation during both homeostasis and the regeneration process [89]. To date, research efforts on insect midgut biology have been limited, with a predominant focus on the model insect *Drosophila*. The midgut epithelium of *Drosophila* experiences complete regeneration approximately every one to two weeks, a process facilitated by the activity of differentiating intestinal stem cells (ISCs) located within the midgut [116]. The *Drosophila* midgut epithelium is mainly comprised of four types of distinct cells: intestinal stem cells (ISCs), undifferentiated progenitor cells known as enteroblasts (EB), specialized absorptive enterocytes (EC), and secretory enteroendocrine cells (EE) [117]. Each cell type has its own unique specialized function. Enterocytes (EC) are the main cells responsible for the production of digestive enzymes and absorption of nutrients. Enteroendocrine cells (EE) serve as chemosensory cells, regulating responses to food, nutrients, and metabolites through the production of neuropeptides and peptide hormones. Intestinal stem cells (ISC) play an important role in the midgut’s regeneration and maintenance. ISCs can divide symmetrically or asymmetrically, leading to the formation of new stem cells or differentiated daughter cells, a process essential for midgut tissue homeostasis during growth, development, or injury [117,118]. Transcriptomes captured by previous studies have uncovered fundamental differences between the cell-types within and between regions of the midgut. However, RNA-seq of specific regions and RNA-seq of FACS cell types have limitations. The diversity and regional differences in the gut make it difficult to characterize subtypes of cells using RNA-seq [89]. Furthermore, conventional sequencing techniques provide an averaged representation across numerous cells, lacking the capacity to analyze a limited number of cells and lose cellular heterogeneity [4].

To address the limitation of earlier studies, until now only a few studies on the fruit fly and mosquito have explored the midgut at the single-cell level with scRNA-seq. Guo and his team [88] for the first time conducted a study on the fly gut employing scRNA-seq, and identified 10 major enteroendocrine (EE) subtypes within the fly gut. These collectively produced approximately 14 different classes of peptide hormones, with an average co-production of two to five distinct classes of peptide hormones per subtype. Additionally, the researchers found that transcription factors such as Mirr and Ptx1 were defining regional EE identities resulting in the specification of two major EE subclasses. The next year, in 2020, Hung and his group accomplished the remarkable achievement of presenting the first cell atlas of the adult *Drosophila* midgut through the application of scRNA-Seq [89]. They reported 22 distinct groups, including stem cells, enteroblasts, enteroendocrine cells (EEs), and enterocytes. In addition, gene expression signatures of these different cell types were identified and specific marker genes assigned to each type of cell. They also identified Cell type-specific organelle features, regional differences among ECs, a transitional state of premature ECs, transcriptome differences between ISCs and EBs, 5 five additional gut hormones, diverse hormone expression of EEs, and paracrine function of EEs. An analysis based on single-nucleus RNA sequencing (snRNA-Seq) was also conducted to investigate the cellular diversity of the midgut and cellular responses to blood meal ingestion for digestion facilitation. The study revealed the presence of 20 distinguishable cell-type clusters in the female midgut of *Aedes aegypti*. These cell types included intestinal stem cells (ISC), enteroblasts (EB), differentiating EB (dEB), enteroendocrine cells (EE), enterocytes (EC), EC-like cells, cardia cells, and visceral muscle (VM) cells. Upon blood meal ingestion, there were significant changes in the overall midgut cell type composition, with a notable increase in the proportions of ISC and three EC/EC-like clusters. Additionally, the transcriptional profiles of all cell types were substantially affected, with significant upregulation of genes involved in various metabolic processes [109]. This technology should be applied to other insect taxa to uncover the mechanisms of insect digestion and cell types involved in the process, as well as gut microbiota that can contribute significantly to the hosts’ performance, such as host nutrition, physiology, and behavior [119,120,121,122]. Further, it is recommended that studies should be carried out on different types of feed, uptake of pesticides, and the interaction of gut microbiota with the host or other environmental pollutants to unveil the cellular response of the insect midgut to a variety of foods and chemicals at single-cell level.

### 3.4. Application in the Insect Reproductive System

Single-cell transcriptomic studies related to the reproductive system of insects have mainly been conducted in the ovary and testis of *Drosophila*. In the *Drosophila* testis, scRNA-seq studies on rare germline stem cells have revealed the reconstruction of developmental trajectories of germ cells during spermatogenesis [98]. Recent research of Witt et al. [96] which compared early germ cells from old and young flies further characterized late spermatogenesis as a source of evolutionary innovation. scRNA-seq was also employed to uncover the of biotic (*Wolbachia*) and abiotic (Antimony) effects on the testis at single-cell level, including changes of the proportion of different types of germ cells and multiple metabolic pathways in germ cells [44,50].

The *Drosophila* ovary serves as a model system for a wide range of research in stem cell biology, organogenesis, and disease regulation. scRNA-seq of adult *Drosophila* ovaries has revealed transcriptomes of rare germline stem cells and consequently uncovered unanticipated cell type complexity in *Drosophila* ovaries [84,86,98]. Further, pseudotime analysis has revealed germ cell trajectory with three branches, one representing early stages of differentiation and the other two representing the paths to oocytes and nurse cells [86,87]. scRNA-seq of larval ovaries contributed to the recovery of a novel cell type corresponding to the elusive follicle stem cell precursors [49], as well as the identification of genes that control early germline cell differentiation by coordinating germline stem cell division with nutritional status [45]. Furthermore, Dong et al. [85] investigated the organization and regulation of stem cell populations in adult tissues, particularly in *Drosophila* ovarian stem cells (FSCs). They revealed spatially distinct clusters representing anterior and posterior escort cells, FSCs, and early follicle cells, which improves our understanding of the regulatory interactions that control FSC behavior.

Using the *Drosophila* ovary as a model system, these studies enhance our knowledge and understanding of organ development, stem cell biology, and disease regulation. In addition, these studies also provide valuable resources and insights into the molecular mechanisms underlying ovarian biology, offering a basis for further research in the field. To our knowledge, scRNA-seq technology has not yet been applied to the reproductive organs of other insect taxa. Therefore, in future this should be taken into consideration.

### 3.5. Application in the Insect Immune System

As an important system for insects to perform immune response, the immune system plays an important role in resisting the invasion of external pathogens. Unlike relatively well-characterized humoral immunity, cellular immunity in insects remains more enigmatic [123]. Only a few studies on the fruit fly, mosquito, and silkworm have taken insect immunity to the single-cell level with scRNA-seq.

In *D. melanogaster*, hemocytes of larvae have been mapped at the transcriptomic level across different inflammatory conditions which revealed the high heterogeneity of plasmatocytes [83], which was consistent with three other independent studies [55,82,100]. Using scRNA-seq, Leitao et al. [56] found that *D. melanogaster* populations evolved immune resistance by differentiating precursors of specialized immune cells called lamellocytes, which resulted in constitutive upregulation of immune-inducible genes when facing a high rate of parasitism by parasitoid wasps. Severo et al. [124] identified two distinct cell populations which resemble granulocytes and oenocytoids in mosquito’s blood cells by scRNA-seq. Single-cell transcriptome of hemocytes of *Anopheles gambiae* and *Aedes aegypti* revealed the functional diversity of hemocytes, and cellular events that underpin mosquito immunity to malaria infection, as well as a previously unidentified cell type, megacyte, in *An. Gambiae* [110]. Besides dipteral insects, scRNA-seq has also been employed in lepidopteran insects like *Bombyx mori*. Feng et al. [125] studied the host responses of different hemocyte clusters in silkworm after infection of BmNPV with single-cell transcriptomics and uncovered great changes in the distribution of hemocyte types caused by BmNPV infection. In their following research, 13 distinct cell clusters of hemocytes in larvae of the silkworm were discussed in detail. Their analysis confirmed the broad division of hemocytes in granulocytes, plasmatocytes, and oenocytoids [108].

### 3.6. Application in the Insect Exocrine System

Recently, single-cell RNA sequencing (scRNA-seq) has been used in only two studies to gain a deeper understanding of insect exocrine glands. The first study examined the remarkable silk gland of the domesticated silkworm *Bombyx mori*, known for its exceptional silk production. Using scRNA-seq, the researchers generated a comprehensive cell atlas of the silk gland, consisting of 14,972 high-quality cells representing 10 different cell types at three early stages of development. Furthermore, they also decoded the developmental trajectory and gene expression patterns of these cells, which provides insight into the regulation of silk gland development and silk protein synthesis [107]. The second study delved into the evolution of a rove beetle gland responsible for secreting a defensive cocktail. It revealed the cooperative evolution of two cell types, each responsible for manufacturing distinct compounds. This coevolution resulted in the emergence of a potent secretion with adaptive value. This research reveals how the interaction between various cell types leads to the development of novel organ-level behaviors [111].

Although these two studies have provided important insights into insect exocrine glands, there is much room for further research in this area. Future research could focus on various other exocrine and endocrine glands or other insect systems to investigate their diverse functions and adaptations. Some possible areas of investigation include: pheromone glands, bee venom glands, reproductive glands, firefly light organs, wax-producing glands etc.

## 4. Challenges and Future Directions (Outlook)

scRNA-seq is a revolutionary technique that provides a deeper understanding of how individual cells function [108]. Unlike conventional high-throughput sequencing, scRNA-seq faces unique challenges, such as **cell isolation:** researchers need to find and capture only a single viable cell: cell dissociation can be a harsh process that can lead to cellular stress and ultimately to transcriptional changes. To minimize this stress, it is recommended that dissociation enzymes (trypsin, collagenase, papain, liberase, and elastase used for insect tissues) be used in combination to enhance the dissociation process so that more viable cells can ultimately be captured. In some insect tissues, such as antenna, wing and body, cells are strongly associated with the cuticle and can be challenging to dissociate [37]. In such cases single-cell nucleus sequencing can be a great opportunity to overcome this problem. Using single-cell nucleus sequencing has a number of advantages: (1) tissues can be stored for long time; (2) it minimizes bias introduced during tissue dissociation step; (3) large and delicate cells may face challenges flowing into microfluidics-based channels like those used by 10× Genomics. However, their nuclei should be easily captured [103]. This technology has been applied successfully to *Drosophila* antenna [94] and abdominal tissues [103]. **Data management**: sometimes it becomes difficult to manage the large amounts of data generated during sequencing, for example to store, analyze, and integrate these data into a coherent system [10]. **Enhancing data analysis efficiency:** there is a need to increase the efficiency of data analysis, reduce costs, gain information about the precise location of cells, and relate the variations revealed by single-cell sequencing to the functions of those cells. Since single-cell research in insects is still in its infancy, the understanding of many species and tissue cell types is not yet perfect, information on the corresponding marker genes is lacking, and cell annotation is incomplete, making subsequent analysis difficult [123].

In recent years, there has been a significant increase in the number of publications using scRNA-seq technology in insect research. This increase indicates the growing acceptance and adoption of this advanced technology by the insect science research community. Most scRNA-seq studies have focused on the model insect *Drosophila*, while many other insect species remain largely unexplored in this research area (Table 1). In the future, comparing single-cell transcriptomics data from different insect species could help us find new cell types and understand cell type specificity of a complex trait. Furthermore, in *Drosophila*, the most extensively investigated system is the nervous system, encompassing studies of the brain, the ventral nerve cord, and optic lobes (Figure 3).

These areas account for approximately 35% of published articles, with the reproductive organs being the second most studied. However, it is worth noting that many other organs remain relatively underexplored (Figure 3, Table 1). To advance our understanding of the underlying mechanisms governing cellular diversity and the specificity of complex traits, it is recommended that this technology be applied to a broader range of organ tissues. This expansion of research would provide valuable insights. Some tissues, such as the wing, leg, and abdominal tissues, may present challenges due to their difficulty in dissociation. To address this issue effectively, a combination of techniques should be employed, and the use of single-cell nuclei analysis could prove to be an appropriate choice [94].

Moreover, scRNA-seq should be combined with other single-cell omics for more comprehensive and detailed understanding of a specific mechanism underlying a specific phenomenon [37]. scRNA-seq lacks spatial information, i.e., it cannot accurately determine the location of cells in tissue. In addition, scRNA-seq cannot reveal epigenetic landscapes. Protein expression also cannot be directly quantified, especially when post-transcriptional changes affect translational efficiency. Recently, several methods have been developed to capture spatial transcriptomes at single-cell resolution, including FISSEQ [126], MERFISH [127], seqFISH [128], and STARmap [129]. Due to the development of single-cell ATAC-seq [130,131], researchers can now quantify chromatin accessibility at the genome level of individual cells. REAP -the ATAC-seq method developed by Peterson et al. [132] allows simultaneous quantification of mRNAs and specific proteins in individual cells using barcoded antibodies. Recently, tissue-specific proteome profiling techniques have been used in *Drosophila*, providing important data that transcriptome analysis cannot reveal [133]. The integration of scRNA-seq with the mentioned single-cell methods helps us get a clear picture of insect cells as well as offering insights into diverse biological systems. Furthermore, when combined with multi-omics methods and functional research we can expect to uncover additional levels of complexity within these systems, opening up the possibility for novel treatment strategies and novel bioengineering applications.

## 5. Conclusions

In summary, the future of single-cell RNA sequencing has immense potential for improving our understanding of cell biology and its applications in various fields. However, overcoming the challenges involved, from the complexity of data analysis to ethical considerations, will be critical to fully realizing the potential of this transformative technology. Despite the fact that insect cells are smaller than those of bigger animals, making it difficult to apply scRNA-seq on them, recent studies using scRNA-seq in insects, especially *Drosophila,* have given us many insights. We have learned about embryo development, brain aging, and germ cell development, and even how diseases begin. Integration of scRNA-seq with other single-cell omics has the potential to unveil exciting new discoveries in the coming decade, especially in insect science.

## Figures and Tables

**Figure 1 cells-13-00091-f001:**
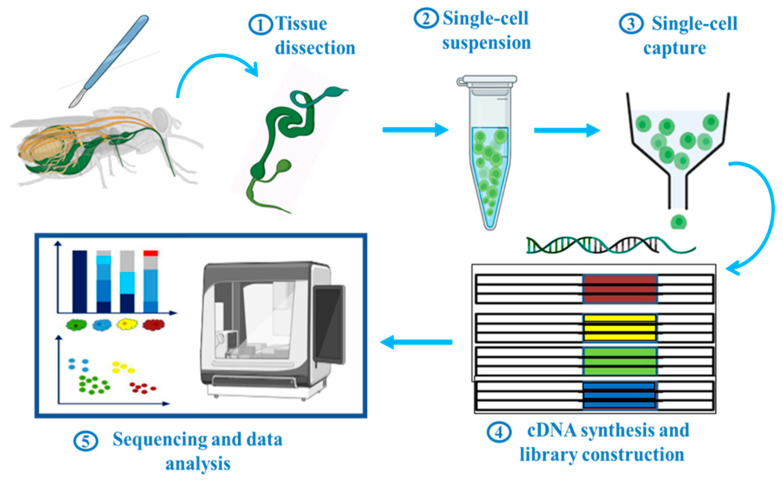
Schematic workflow of scRNA-seq in insects. Generally, it includes five main steps: Tissue dissection, single-cell suspension, single-cell capture, library construction, sequencing and data analysis. This figure was created using BioRender (www.biorender.com; accessed on 1 August 2023).

**Figure 2 cells-13-00091-f002:**
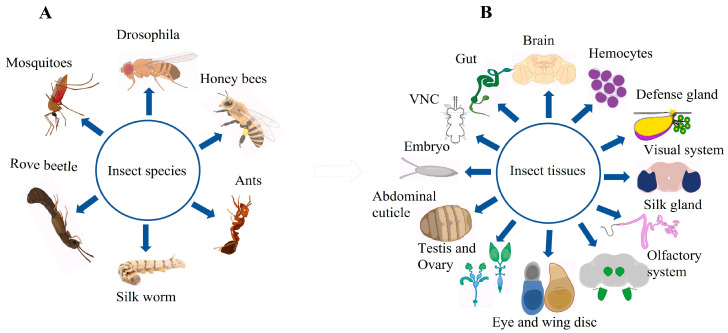
Overview of various insects and their organs studied using single-cell RNA sequencing (scRNA-seq). This figure illustrates Panel: (**A**), Different insects on which scRNA-seq technology has been applied; Panel: (**B**), organs or tissues of insects on which scRNA-seq technology has been applied for comprehensive gene expression profiling (see Table 1 for relevant references and description). This figure was created using BioRender (www.biorender.com; accessed on 12 August 2023).

**Figure 3 cells-13-00091-f003:**
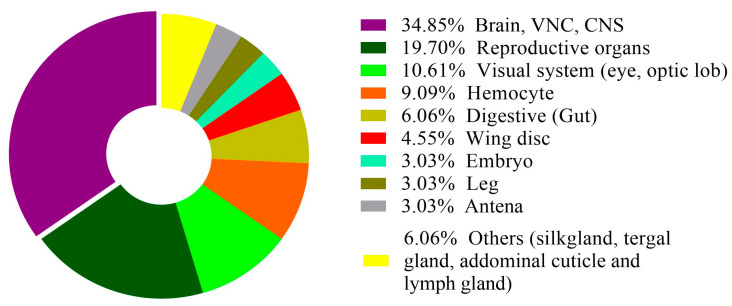
The color code represents the proportion of articles related to each tissue category within the dataset.

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
