# Peer review of "Insect Insights at the Single-Cell Level: Technologies and Applications"

_cells, 2023, doi:10.3390/cells13010091_

Round 1
Reviewer 1 Report
Comments and Suggestions for Authors
This overall well written review could be improved by avoiding general phrases such as "open new door", "great discoveries", "promoted understanding, "unique resources", "shed light" which do not actually provide any information regarding the mentioned studies. Some examples are in lines 104, 108, 154, 158-159, 190- 192 and throughout the text. Instead, a brief description of findings and not just naming what was studied would be helpful. Otherwise the reader would still need to read the original papers to understand what were the main findings. Examples of very nicely described studies are lines 194-198, 235-264, 221-234 (even though it is not clear if the findings in lines 221-234 were from scRNA-seq studies or other methods, clarification would help).
Author Response
Thank you for your valuable feedback. We appreciate your insightful comments regarding the use of general phrases in our manuscript. Line 104 has been changed as suggested (revised version line 130). General phrase has been removed and more information have provided. We appreciate your keen observation regarding the general phrases used in line 108 and completely agree on the importance of avoiding ambiguity. In response to your comment, we want to assure you that the studies mentioned in line 108 are thoroughly explained in subsequent sections of the manuscript. For example, papers cited on intestine are further explained in section 3.3 “application in insect digestive system”. Similarly, brain aging and neural development related articles are further explained in section 3.2 “application in insect nervous system”. We are worrying if we added more information about the cited articled here in this part then the introduction will be too lengthy. In line 154, 158-159 more information has been added in the revised version (revised version line 192, 197). Suggested changes have been done in line 190-192, a brief description of the mentioned study have been added in the revised version (line 243-246). Findings in Line 221-234 are not from the scRNA-seq studies; actually, these lines provide basic information about the cell types of the midgut.
Reviewer 2 Report
Comments and Suggestions for Authors
This study offers a thorough examination, presenting an overview of techniques and applications of single-cell RNA sequencing (scRNA-seq) in the context of insects. The focus is particularly directed towards recent studies that explore various tissues of insects. The review is well-constructed and serves as a valuable resource for individuals engaged in insect studies, providing insights into the advancements in scRNA-seq within this field. However, one noteworthy observation is that the authors rarely introduce quality of the data presented in Table 1. Enhancing the discussion on data quality, including insights into the quality control (QC) processes employed, would greatly benefit readers. This additional information could provide readers with a better understanding of whether their own studies align with established quality standards.
Author Response
Thank you for your thorough review of our manuscript. We appreciate your valuable feedback, and we have addressed these corrections as suggested. we have added a paragraph (revised version: line 102-118) on the quality control (QC) in section 2 of the manuscript (The effectiveness and overview of the single-cell RNA-sequencing workflow). Besides for more detailed knowledge on the QC, we cited other reviews in the paragraph which will benefit the readers too.
Reviewer 3 Report
Comments and Suggestions for Authors
In their manuscript titled “Insect Insights at the Single-Cell Level: Technologies and Applications,” Sun, Shao and Iqbal present a review of single-cell sequencing studies from D. melanogaster and other insects. They discuss major conclusions from these studies, as well as presenting a comprehensive table of scSeq studies in insects to date. The review ends with suggestions for future studies and discussion of challenges and limitations of current single cell methods. The topic is relevant and the compilation of the existing literature is useful. The review should be of interest for the Drosophila community, as well as scientists studying other insects. The review will also be useful for vertebrate scientists looking for possible evolutionary links or technical applications of single-cell sequencing.
I have several recommendations to improve the manuscript.
Although the overall organization and flow of the manuscript, as well as the clarity of ideas, is nicely presented, the manuscript needs English and grammar proof reading. There are many grammar mistakes and wording oddities throughout the manuscript, including word choice, verb tense, phrasing, article use, capitalization, etc.
There are multiple paragraphs in the manuscript where citations need to be added as appropriate. A summary of all locations that need citations added include:
· lines 59-85 – citations missing
· Lines 212-219 – citations missing
· Lines 340-347 – citations missing (esp. sentences lines 346 & 348)
· Section 4 – please take a look at and add citations throughout this part, some paragraphs have citations, others do not.
Section 2: Section title states “Effectiveness and overview.” The section does provide an overview, but there is only limited discussion comparing the different methods for seq. There is also no discussion of “effectiveness”, for example cell-type representation, biased recovery of cell types, depth of sequencing, etc. The authors could add a table, for example, summarizing basic approach as well as the advantages versus disadvantages of the different methods. As it is, the manuscript just lists them and doesn’t provide any detail. The authors could alternatively point the reader to other reviews that cover technique details and approaches.
Were the figures generated with BioRender? If so, this should be cited in the legend. The figures themselves look professional, are easy to understand, and compliment the text nicely.
Figure 2. Should include the genus species of the insects studied. Would probably be good to have A and B panel designations, or to refer to panel on the left versus panel on the right in the legend for clarity. The citations for relevant studies should be included in this figure or alternatively in the legend there should be a direct reference to Table 1 to clarify that the relevant references and description of all relevant studies are available in Table 1.
Section 3.1 – The authors should mention the Fly Cell Atlas in this section. There are multiple single-cell studies focused on a single tissue or organ, but conceptually I usually reference the Fly Cell Atlas to get an overview, and then go into papers on individual tissues for greater resolution.
Line 178 is unclear, it seems to state that scSeq has only been applied to the visual system, but the preceding paragraph discussed the olfactory system, demonstrating general application to the CNS.
Line 196 is rather vague, and doesn’t identify the relevant transcription factors.
Also, Section 3.3 has a two-paragraph introduction to the system, which is different than the lay-out of the other sections in the review. The two-paragraph introduction (lines212-233) could more clearly point out the limitations of previous studies that were addressed by scSeq, as suggested in line 235.
Paragraph lines 340-357. The use of first, second and third in this paragraph is a bit odd, as there are multiple limitations discussed under each numeric designation. Use of a different connector or a topic/theme rather than ordinals first, second and third would be more appropriate. The limitations discussed are very relevant. The single nucleus seq, for example, could have another sentence or two for explanation, as could several of the other limitations. This is an important part of the review, as these are critical parameters for design of scSeq experiments that should be considered when planning new experiments.
Minor corrections:
Line 70 should start a new paragraph.
Line 262 is in the first person, should be in third.
Table 1. This table is a useful reference. Column 1 – rather than Drosophila, D. melanogaster would be a better label, as there are a wide range of Drosophilids which may be used for single cell studies at some point. Short-hand abbreviations (i.e. 10x, Drosophila) should be clarified in the table legend.
Comments on the Quality of English Language
Although the overall organization and flow of the manuscript, as well as the clarity of ideas, is nicely presented, the manuscript needs English and grammar proof reading. There are many grammar mistakes and wording oddities throughout the manuscript, including word choice, spelling, verb tense, phrasing, article use, capitalization, etc. (too numerous to point out one by one in this review). At points, the grammar mistakes do detract from the ease of reading, and the manuscript should be proof-read prior to publication.
Author Response
Although the overall organization and flow of the manuscript, as well as the clarity of ideas, is nicely presented, the manuscript needs English and grammar proof reading. There are many grammar mistakes and wording oddities throughout the manuscript, including word choice, verb tense, phrasing, article use, capitalization, etc.
Response: Thank you for providing us with your detailed comments and insightful suggestions on our manuscript. We thoroughly reevaluated the relevant sections and made the necessary improvements to ensure that the manuscript meets the highest standards of clarity and coherence.
There are multiple paragraphs in the manuscript where citations need to be added as appropriate. A summary of all locations that need citations added include:
- lines 59-85 – citations missing
- Lines 212-219 – citations missing
- Lines 340-347 – citations missing (esp. sentences lines 346 & 348)
- Section 4 – please take a look at and add citations throughout this part, some paragraphs have citations, others do not.
Response: Thank you very much for pointing out our mistake. We are sorry for this. From 59-85 some citations have been added to the revised manuscript and at the end of the paragraph we suggested some other reviews for more compressive knowledge of the mentioned steps. In line 212-219, line 340-347 and in section 4, citations have been provided where needed.
Section 2: Section title states “Effectiveness and overview.” The section does provide an overview, but there is only limited discussion comparing the different methods for seq. There is also no discussion of “effectiveness”, for example cell-type representation, biased recovery of cell types, depth of sequencing, etc. The authors could add a table, for example, summarizing basic approach as well as the advantages versus disadvantages of the different methods. As it is, the manuscript just lists them and doesn’t provide any detail. The authors could alternatively point the reader to other reviews that cover technique details and approaches.
Response: Thank you very much for your valuable feedback on Section 2. We acknowledge the suggestion for a table summarizing basic approaches. The reason we didn't include a table in the manuscript is because other reviews summarizing the methods and application of scRNA-seq have already provided more detailed information on basic approach as well as the advantages versus disadvantages of the different methods. In the revised version (lines 89-95), we added some information about the distinctive features of these approaches. Additionally, as suggested by the reviewer, we have directed readers to other reviews for a more comprehensive understanding of the different methods.
Were the figures generated with BioRender? If so, this should be cited in the legend. The figures themselves look professional, are easy to understand, and compliment the text nicely.
Response: Thank you very much. It has been cited. This figure was created using BioRender (www.biorender.com)
Figure 2. Should include the genus species of the insects studied. Would probably be good to have A and B panel designations, or to refer to panel on the left versus panel on the right in the legend for clarity. The citations for relevant studies should be included in this figure or alternatively in the legend there should be a direct reference to Table 1 to clarify that the relevant references and description of all relevant studies are available in Table 1.
Response: Thank you very much for your guidance. Figure 2 has been revised accordingly. We are worrying if we provide Genus and species name to the figure then the figure will have too much text and will be difficult for the readers to understand. To overcome this, we cited table 1 in the legend of figure 2, and made some changes in the column 1 of the table 1 (species and genus names are added in table 1, column 1). Besides, we added panel A and panel B. Panels: A, Different insects on which scRNA-seq technology was applied. Panels: B, Organs or tissues of insects on which scRNA-seq technology has been applied for comprehensive gene expression profiling.
Section 3.1 –
Line 178 is unclear, it seems to state that scSeq has only been applied to the visual system, but the preceding paragraph discussed the olfactory system, demonstrating general application to the CNS.
Response: Thank you very much for your correction. We are sorry for this mistake. It was some grammatical mistake. We intended to express that “scSeq has been applied to the visual system of only Drosophila, instead of other insects”. In order to avoid such misunderstanding, certain sentences have been deleted from that part. Besides we also made some changes in line 178 (revised version line 237). Some modifications have been made at the start sentence to make it easier for the readers to understand. Now next, the paragraph explains the optic lobes, primarily associated with the processing of visual information.
Line 196 is rather vague, and doesn’t identify the relevant transcription factors.
Response: Thank you for pointing out this mistake. This line has been revised. (Line 246 in the revised version)
Also, Section 3.3 has a two-paragraph introduction to the system, which is different than the lay-out of the other sections in the review. The two-paragraph introduction (lines212-233) could more clearly point out the limitations of previous studies that were addressed by scSeq, as suggested in line 235.
Response: Thank you very much for your time and effort and pointing out our mistakes. We combined these two paragraphs to make it similar to the layout of the other sections. Besides we also added a small paragraph on the limitations of previous studies (Revised version Line 292-298).
Paragraph lines 340-357. The use of first, second and third in this paragraph is a bit odd, as there are multiple limitations discussed under each numeric designation. Use of a different connector or a topic/theme rather than ordinals first, second and third would be more appropriate. The limitations discussed are very relevant. The single nucleus seq, for example, could have another sentence or two for explanation, as could several of the other limitations. This is an important part of the review, as these are critical parameters for design of scSeq experiments that should be considered when planning new experiments.
Response: To address the above-mentioned shortcomings and improve the overall quality of the manuscript, we have made the suggested changes in the lines 340-357 (see revised version Line 406-422).
We replaced each numeric designation with a short title as suggested. First is replaced with cell isolation, second is replaced with data management while third numeric is replaced with enhancing data analysis and efficiency.
Besides we also provided more information on snRNA-seq. the changes are made in line- -to- in the revised version (Revised version line 414-418).
Minor corrections:
Line 70 should start a new paragraph.
Response: Now in the revised version, new paragraph starts from line 70.
Line 262 is in the first person, should be in third.
Response: We are sorry for this mistake. It has been revised
Table 1. This table is a useful reference. Column 1 – rather than Drosophila, D. melanogaster would be a better label, as there are a wide range of Drosophilids which may be used for single cell studies at some point. Short-hand abbreviations (i.e. 10x, Drosophila) should be clarified in the table legend.
Response: Thank you very much for your deep insight. Column 1 in the table has been changed as suggested besides a legend is also provided to the table explaining the abbreviations in the table.
Although the overall organization and flow of the manuscript, as well as the clarity of ideas, is nicely presented, the manuscript needs English and grammar proof reading. There are many grammar mistakes and wording oddities throughout the manuscript, including word choice, spelling, verb tense, phrasing, article use, capitalization, etc. (too numerous to point out one by one in this review). At points, the grammar mistakes do detract from the ease of reading, and the manuscript should be proof-read prior to publication.
Response: We carefully checked the spelling and grammar mistakes.
Round 2
Reviewer 2 Report
Comments and Suggestions for Authors
Thanks for revising. The authors have addressed my concerns.